# Establishment and Validation of Predictive Model of Tophus in Gout Patients

**DOI:** 10.3390/jcm12051755

**Published:** 2023-02-22

**Authors:** Tianyi Lei, Jianwei Guo, Peng Wang, Zeng Zhang, Shaowei Niu, Quanbo Zhang, Yufeng Qing

**Affiliations:** 1Department of Rheumatology and Immunology, Affiliated Hospital of North Sichuan Medical College, Nanchong 637000, China; 2Research Center of Hyperuricemia and Gout, Affiliated Hospital of North Sichuan Medical College, Nanchong 637000, China; 3Department of Geriatrics, Affiliated Hospital of North Sichuan Medical College, Nanchong 637000, China; 4Department of Infection, Affiliated Hospital of North Sichuan Medical College, Nanchong 637000, China

**Keywords:** gout, tophus, machine learning, prediction model, Shapley Additive exPlanations (SHAP)

## Abstract

(1) Background: A tophus is a clinical manifestation of advanced gout, and in some patients could lead to joint deformities, fractures, and even serious complications in unusual sites. Therefore, to explore the factors related to the occurrence of tophi and establish a prediction model is clinically significant. (2) Objective: to study the occurrence of tophi in patients with gout and to construct a predictive model to evaluate its predictive efficacy. (3) Methods: The clinical data of 702 gout patients were analyzed by using cross-sectional data of North Sichuan Medical College. The least absolute shrinkage and selection operator (LASSO) and multivariate logistic regression were used to analyze predictors. Multiple machine learning (ML) classification models are integrated to analyze and identify the optimal model, and Shapley Additive exPlanations (SHAP) interpretation was developed for personalized risk assessment. (4) Results: Compliance of urate-lowering therapy (ULT), Body Mass Index (BMI), course of disease, annual attack frequency, polyjoint involvement, history of drinking, family history of gout, estimated glomerular filtration rate (eGFR), and erythrocyte sedimentation rate (ESR) were the predictors of the occurrence of tophi. Logistic classification model was the optimal model, test set area under curve (AUC) (95% confidence interval, CI): 0.888 (0.839–0.937), accuracy: 0.763, sensitivity: 0.852, and specificity: 0.803. (5) Conclusions: We constructed a logistic regression model and explained it with the SHAP method, providing evidence for preventing tophus and guidance for individual treatment of different patients.

## 1. Introduction

Gout is an inflammatory disease caused by the deposition of monosodium urate (MSU) crystals in joint and non-joint structures [1]. Patients with gout experience a variety of symptoms, including severe pain, acute and persistent inflammatory arthritis, and symptoms associated with chronic disease [2]. As gout progresses, clinical symptoms of advanced disease characterized by a tophus may appear, primarily the recurrence of chronic granulomas resulting from a continuous deposition of MSU [3,4]. Formation of a tophus can lead to joint deformities and associated joint injury, fracture and skin rupture, or infection [5,6]. In addition, tophi can occur in unusual sites (such as the heart valve, carpal canal, larynx, and spine) and cause complications. Growing studies indicate that multiple factors may influence the development of tophi, such as course of disease, estimated glomerular filtration rate (eGFR), compliance of urate-lowering therapy (ULT), etc. [7,8]. The main treatment for tophi is pharmaceutical intervention including purine and non-purine xanthine oxidase inhibitors, uric acid excretion agents, uric acid enzymes, and whole-human anti-IL-1β monoclonal antibodies, as well as other interventions such as surgical removal [9,10,11,12]. However, if the current treatment regimen is not effective, the presence of a tophus can lead to significant complications. Therefore, an early detection of risk factors and establishment of a prediction model has great significance to improve early prevention of tophi. Machine learning (ML) is an emerging field of medicine that represents a powerful set of algorithms capable of representing, adapting, learning, predicting, and analyzing data; moreover, ML is considered as the future of biomedical research, personalized medicine, and computer-aided diagnosis [13,14]. Therefore, this study used a variety of ML classification models to build a prediction model [15,16]. Through collecting and sorting the clinical data of gout patients, influencing factors of tophus were analyzed to provide clinical evidence for early treatment of tophus formation. The application of the ML model has shown accurate individual prediction and promising clinical application prospects [17]. However, its application in real clinical practice and interpretable evidence for risk prediction models are limited [18,19]; therefore, we also used the Shapley Additive exPlanations (SHAP) interpretation tool to provide an intuitive explanation of risk leading to patient predictions [20]. The tool can generate individual probabilities of clinical events by integrating determinants, and it also meets the need for a combination of biological and clinical models, contributing to the development of personalized medicine. The aim of this study is to establish a more suitable clinical study of tophi in gout patients, and establish a corresponding predictive model. It is helpful to improve the diagnosis system of the tophus and provide more reference for clinicians.

## 2. Materials and Methods

### 2.1. Materials

#### 2.1.1. Subjects

A total of 792 gout patients from 1 January 2018 to 30 June 2022 attended Rheumatic Immunity Department, Affiliated Hospital of North Sichuan Medical College.

#### 2.1.2. Inclusion Criteria

Inclusion criteria are as follows: (1) compliance with the diagnostic criteria for gout established by ACR/EULAR in 2015; (2) informed consent and voluntary participation in the study; and (3) having complete clinical data.

#### 2.1.3. Exclusion Standards

Exclusion criteria are as follows: (1) those with serious diseases such as chronic cardiac insufficiency, liver diseases, malignant tumors, and mental diseases; (2) intake of certain drugs (such as diuretics, aspirin, cytotoxic drugs, antituberculosis drugs, etc.); and (3) patients who were unable to cooperate, unwilling to participate, or whose clinical data were incomplete.

### 2.2. Methods

#### 2.2.1. Grouping Methods and Diagnosis of Tophi

All the patients were divided into two groups. Diagnosis of tophi: Light yellow or white uplift or neoplasm of various sizes, hard subcutaneous lump, or/and color, Doppler ultrasonography showing evidence of dual-track sign, tophus, bone erosion, or/and X-ray computed tomography showing high-density lump. These were comprehensively evaluated in combination with patients’ clinical histories.

#### 2.2.2. Study Indicators

There were 43 variables: (1) General data, including tophus, sex, compliance of ULT, Body Mass Index (BMI), course of disease, annual attack frequency, polyjoint involvement, history of drinking, history of smoking, history of hypertension, history of high altitude residence, history of high purine diet, history of sugary diet, history of diabetes, history of hyperlipidemia, history of kidney stones, history of kidney crystallization, family history of gout. (2) Laboratory examination, including eGFR, Erythrocyte sedimentation rate (ESR), White cell rate (WBC), Granulocte (GR), Lymphocyte (LY), Monocyte (MO), Red blood cell (RBC), Hemoglobin (HGB), Hematocrit (HCT), Mean Corpuscular Volume (MCV), Mean corpuscular hemoglobin concentration (MCHC), Platelet (PLT), Mean platelet volume (MPV), Plateletcrit (PCT), Platelet distribution width (PDW), Uric acid (UA), Urea, Creatinine (Crea), Alanine aminotransferase (ALT), Aspartate aminotransferase (AST), serum albumin (ALB), globulin (GLOB), Cystatin C (CysC), Urine Ph, urine specific gravity.

#### 2.2.3. Construction and Evaluation of Predictive Models

After selecting characteristic factors from all independent variables, we divided gout patients into training set and testing set. Multiple ML classification models were applied for comprehensive analysis, comparison on the importance of each index in training set and testing set of different models. Furthermore, we utilized the optimal model to evaluate and verify the results. The SHAP presentation model as a whole and single sample interpretation were also developed. Detailed steps were as follows: (1) Screening characteristic factors: First, R software (glmnet4.1.2) was used to conduct the least absolute shrinkage and selection operator (LASSO) regression analysis and adjusting the variable screening and complexity. Then, LASSO regression analysis results were used to conduct multifactor logistic regression analysis with SPSS, and finally, we obtained the characteristic factors of *p* < 0.05. (2) Data division: Pyskthon (0.22.1) random number method was used to randomly divide the gout patients into training set and test set according to the ratio of 7:3, of which 491 were in the training set and 211 were in the testing set. (3) Classified multi-model comprehensive analysis: eXtreme Gradient Boosting (XGBoost), Logistic regression, Light Gradient Boosting Machine (LightGBM), RandomForest, Adaptive Boostint (AdBoost), Multilayer Perceptron (MLP), support vector machine (SVM), K-Nearest-Neighbors (KNN), Gaussian Naïve Bayes (GNB) were built by using python (sklearn 0.22.1, xgboost 1.2.1, lightgbm 3.2.1). We then trained and tested the above parameter model (Repeat 10 samples), analyzed the importance of the training set and testing set indicators in different models, and selected the optimal model. Python (sklearn 0.22.1) was used to construct the area under the receiver operating characteristics (ROC) curve and is often used to describe tools for diagnostic testing or the identification accuracy of predictive models [21]. R software (rmda 1.6) was used to plot the decision curve analysis (DCA) that is essentially the decision analysis. Thus, it is possible to decide whether to use one model, or which one of several models was the optimal, with significant advantages in assessing the clinical applicability of the model [22]. Python (sklearn 0.22.1) calibration curves were used to measure the model’s prediction power, and comprehensive assessment of the predictive model was employed to validate its usefulness in decision support or more general simulation modeling [23]. Python (sklearn 0.22.1) was used to plot precision recall (PR) curves, which were widely used to evaluate the performance of models. PR and area under PR (AP) curve can provide a valuable complement to existing model evaluation methods [24]. (4) Training, verification, and testing of the optimal model: the training set was cross-verified with 10 folds and evaluated with the testing set. Python (sklearn 0.22.1) Draw learning curves were used to evaluate the model fit and stability of training and validation sets [25]. (5) Python (shap 0.39.0) was used to draw the SHAP interpretation of importance and contribution to the model and interpret the model results by calculating the contribution of each feature to the predicted results. In addition, the SHAP was built for a single sample and tries to calculate the prediction performance [26].

## 3. Statistical Analysis

Variables were all included in comparison between training and testing sets. Continuous variables were expressed as median and Inter-Quartile Range (IQR) and compared using the Mann–Whitney U-test. Categorical variables were expressed in number and percentage and compared using chi-square tests. Bilateral *p* values less than 0.05 were considered statistically significant. SPSS (version 25.0), R (version 3.6.1), and Python (version 3.4.3) were used for statistical analysis.

## 4. Results

### 4.1. Comparison of Baseline Data

In this study, we excluded a total of 90 gout patients with other serious diseases, See Appendix A. Regarding the analysis of 702 gout patients, all variables were investigated at the initial diagnosis, and the compliance of ULT was defined as poor compliance if the medication possession ratio (MPR) [27,28,29,30] was lower than 60% and as high compliance if the MPR ≥ 60%. The annual attack frequency could be divided into the severity of at least 12 times per year and less than 6 times per year, with less than 6 as low degree, 6–12 as medium degree, and more than 12 as high degree. History of drinking was defined by no history of drinking, drinking less than 70 g per week as moderate history of drinking, drinking ≥ 70 g per week and drinking years ≥ 10 years as excessive drinking. Polyjoint involvement was defined by the presence of a tophus above three joints. The specific baseline data of the final training set and the test set are shown in Table 1. There was no significant difference between the two groups (*p* > 0.05).

### 4.2. Screening of Characteristic Factors for Risk of Tophi in Gout Patients

LASSO regression analysis was conducted on the remaining independent variables with presence of a tophus as the dependent variable (Figure 1). LASSO can compress variable coefficients to prevent overfitting and solve severe collinearity problems [31]. The results showed that (lambda with minimum mean square error = 0.024) 42 independent variables were reduced to 11, including sex, compliance of ULT, BMI, course of disease, annual attack frequency, history of drinking, family history of gout, polyjoint involvement, eGFR, ESR, and UA. To further control the influence of confounding factors, the above 11 independent variables were analyzed using multivariate logistic regression [32]. Finally, only compliance of ULT, BMI, course of disease, annual attack frequency (>12 times), history of drinking (drinking ≥ 70 g per week/drinking years ≥ 10 years), family history of gout, polyjoint involvement, eGFR, and ESR were determined as characteristic factors (*p* < 0.05), as Table 2.

### 4.3. Comprehensive Analysis of Classified Multi-Model

XGBoost, Logistic, LightGBM, RandomForest, AdaBoost, MLP, SVM, KNN, and GNB were trained and repeated 10 times. The model was evaluated using area under curve (AUC) values [21], and the results indicated that XGBoost, LightGBM, and RandomForest were the highest in the training set and Logistic was the highest in the testing set (Figure 2a,b); see more details in Appendix A. The AUC indicator focuses on the predictive accuracy of the model and does not tell whether the model is clinically usable or which one of the two is more preferable [21,33]. Therefore, the DCA, calibration curves, and PR curve were analyzed. The DCA evaluates Logistic and RandomForest for a better clinical suitability (Figure 2c). Calibration curves showed a higher accuracy of GNB and Logistic model predictions (Figure 2d). In training and test sets, the Logistic model showed the optimal performance, with the highest AP value in the test set (Figure 2e,f). Comprehensive analysis demonstrated that Logistic could be considered the optimal model.

### 4.4. The Best Model Building and Evaluation

Logistic regression analysis and 10-fold cross validation were performed on the training set. The results showed that the average AUC of the training set was 0.876 (0.838–0.914), the average AUC of the verification set was 0.854 (0.733–0.972), and the AUC of the testing set was 0.888 (0.839–0.937) (Figure 3a–c). The AUC of the training set, the verification set, and the testing set was finally stable at about 0.85, and the model prediction effect was accurate. As the performance of the verification set under the AUC index was lower than the test set or the ratio was lower than 10%, the model fitting could be considered successful, and the learning curve indicated that the training set and the verification set had a strong fitting and high stability [25] (Figure 3d). These results indicated that the logistic regression model could be used for the classification modeling task of the data set.

### 4.5. The SHAP to Model Interpretation

To visually explain the selected variables, we used SHAP to illustrate how these variables predicted the formation of a tophus in the model [26]. Figure 4a shows the nine most important features in our model. In each feature important line, the attributions of all patients to the results are plotted with different colored dots, where red dots represent high risk values and blue dots represent low risk values. Decreased BMI and compliance of ULT (MRP < 60%), longer course of disease, high annual attack frequency (>12 times), history of excessive drinking, family history of gout, polyjoint involvement, decreased eGFR, and increased ESR would elevate the formation of tophi in gout patients. Figure 4b shows the ranking of nine risk factors evaluated by the average absolute SHAP value, with the x-axis SHAP value indicating the importance of the forecast model. In addition, we provided two typical examples to illustrate the interpretability of the model, one was a gout patient without a tophus with a low SHAP predictive score (0.133) (Figure 4c), while another gout patient with a tophus had a higher SHAP score (0.722) (Figure 4d).

## 5. Discussion

In this study, we excluded a total of 90 patients; of these patients, only one had heart disease, two had liver damage, and one had lung cancer. The prevalence of tophus was about 4.4%. The risk of hyperuricemia in heart disease is high and may be due to decreased renal perfusion and UA excretion [34,35,36]. Elevated levels of xanthine oxidase (XO) were also reported in patients with heart failure [37]. In addition, some patients with decompensated heart failure (DHF) develop sodium retention that stimulates renal urate anion exchangers that affect UA [38]. Diuretic doses are usually higher than baseline doses, resulting in reduced UA excretion and possible hyperuricemia [39]. The liver is the main site of UA biosynthesis. XO participates in the formation of UA and may releases XO after impaired liver function [40]. Most patients with advanced liver disease have hypoproteinemia. It should be noted that the presence of carboxylic acid groups in albumin is necessary for the positive effect of albumin on MSU nucleation [41]. Chemotherapy in cancer patients can lead to increased cell destruction, significantly raising UA levels, which in turn can lead to gout [42]. In the case of mental illness, this part of the population is excluded because it is unable to provide regular outpatient care. At the same time, gout treatment drugs are harmful to the liver. In our investigation, many of the liver diseases associated with gout patients were not treated properly, which may interfere with our study of tophus formation in gout patients. In addition, there are drugs that can cause hyperuricemia, which can lead to gout symptoms [43]. The gout manifestations in these patients may be transient, so we do not consider them a risk factor for tophi.

Our results show that nine clinical characteristic variables were screened by LASSO and multivariate logistic regression analysis from 42 clinical variables (compliance of ULT, course of disease, polyjoint involvement, history of drinking (drinking ≥ 70 g per week/years of drinking ≥ 10 years), eGFR, annual attack frequency (>12 times), BMI, ESR, and family history of gout to assess the risk of tophi in patients. About 25% of gout patients in our study developed a tophus. Several studies have reported predictive risk factors as the clinical presentation for tophus patients [7,44,45,46,47]. For example, a Chinese retrospective study has shown that disease duration and joint involvement in the upper extremities affected joints and kidney stones, and that hypertension is a risk factor for the development of subcutaneous tophi, while BMI may be a protective factor for tophus [44]. Beilei Lu et al. reported a lower eGFR and a longer disease duration as independent risk factors for tophus formation in gout patients. Double Profile Sonography (DCS) was higher in patients with tophus than those without [7]. Another study has shown that age and DCS are potential risk factors for tophi [45]. A simple study of metabolic markers associated with tophi has shown that UA, eGFR, γ-GT, and ALT are related to tophi, and that the γ-GT/ALT ratio can be used as a predictor or monitor of tophi [47]. A recent study reported that high serum free fatty acid level is independently correlated with risk of tophi, which may promote tophus deposition by lowering urine pH [46].These findings often rely on data labeled by human experts. Despite the differences, it is indicated that course of disease, eGFR, and DCS may play a more significant role. Unfortunately, this study did not include clinical manifestations of joint ultrasound in gout patients. In our study, the course of disease and the role of eGFR in the formation of tophus were consistent.

In addition, we found clinical factors that might influence the formation of tophi, such as compliance of ULT, polyjoint involvement, history of drinking, annual attack frequency, BMI, ESR, and family history of gout. In this study, compliance of ULT was considered the most significant predictor. A multicenter prospective study reported that ultrasound monitored a reduction in urate deposition after ULT in gout [8]. Another prospective study also found a gradual reduction in the size of tophi after lesinurad plus fipronil treatment [48]. The 2012 American Academy of Rheumatology Gout Management Guidelines recommends ULT as an initial treatment for gout with tophi [49]. Reasonable ULT can reduce serum UA level and pathological MSU deposition [50]. Alcohol is an important risk factor for gout. Ethanol consumes ATP, increases lactic acid production, increases UA production, and reduces UA excretion from the kidneys [51]. BMI was positively correlated with body temperature, probably because of the thicker subcutaneous fat tissue and better thermal insulation [52,53]. However, lower temperatures result in lower urate solubility [54]. In a large data analysis, both men and women had U-shaped UA–BMI relationships, which was positively correlated with a BMI of 20 kg/m^2^ and negatively correlated with a BMI of 20 kg/m^2^ [55]. Interestingly, in elderly patients, BMI was positively associated with quadriceps muscle mass [56]. Albumin was positively correlated with muscle mass in males, and negatively correlated with muscle mass in females [57]. Albumin is a large molecule that may increase the solubility of UA [58,59]. Elevated levels of hyaluronic acid in the blood of obese people can lead to a slight increase in urine solubility [60,61]. Another cross-sectional study from China also suggests that BMI may be a protective factor. This evidence suggests that those with lower BMI may be more likely to form tophi [44]. Polyjoint involvement, annual attack frequency, and more ESR may indicate the frequency and severity of acute spasms, reflecting the potential deposition of MSU. At the same time, genetic factors and immune status may also affect the formation of tophi.

Although there are many risk factors for tophi, no predictive model has been established. In this study, we used several ML models, and found that the logistic regression model performed better than other ML models after analyzing the AUC, DCA, calibration curves, and PR curves. However, it has always been a challenge to interpret the ML prediction model more comprehensively and to visually present the predictive results to clinicians. Therefore, we applied the SHAP method to the logistic regression model to achieve the optimal predictive effect and interpretability. We identified some important variables associated with the development of tophi in gout patients.

However, our study has several limitations. Firstly, there was no gold standard inclusion or exclusion criteria for tophi. Secondly, the sample size was relatively small in the study; the data were collected in a single institution, it was not a multi-center study. Therefore, the results were of limited generality. Furthermore, although a high consistency was achieved in the repeatability analysis within the training and testing set, some inevitable errors may occur due to segmentation uncertainty. Finally, the design of the study did not include some variables such as 24 h quantitative UA and joint ultrasound in the analysis. Longitudinal or prospective case controlled studies are also needed to further explain the relationship between risk factors and tophus formation.

## 6. Conclusions

In conclusion, this study constructed a predictive model based on the ML model, and the logistic regression model showed a better performance in this study. In addition, we provided a personalized risk assessment for the development of tophi in gout patients explained by SHAP. This effective computer-aided approach can help first-line clinicians and patients identify and intervene in the occurrence of tophi.

## Figures and Tables

**Figure 1 jcm-12-01755-f001:**
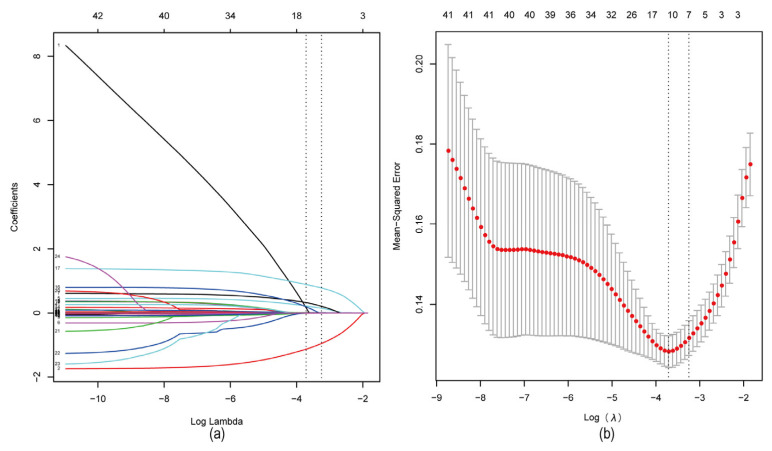
LASSO regression analysis was used to select characteristic factors. (**a**) The use of 10-fold cross-validation to draw vertical lines at selected values, where the optimal lambda produces nine nonzero coefficients. (**b**) In the LASSO model, the coefficient profiles of 42 texture features were drawn from the log (λ) sequence. Vertical dotted lines are drawn at the minimum mean square error (λ = 0.024) and the standard error of the minimum distance (λ = 0.039).

**Figure 2 jcm-12-01755-f002:**
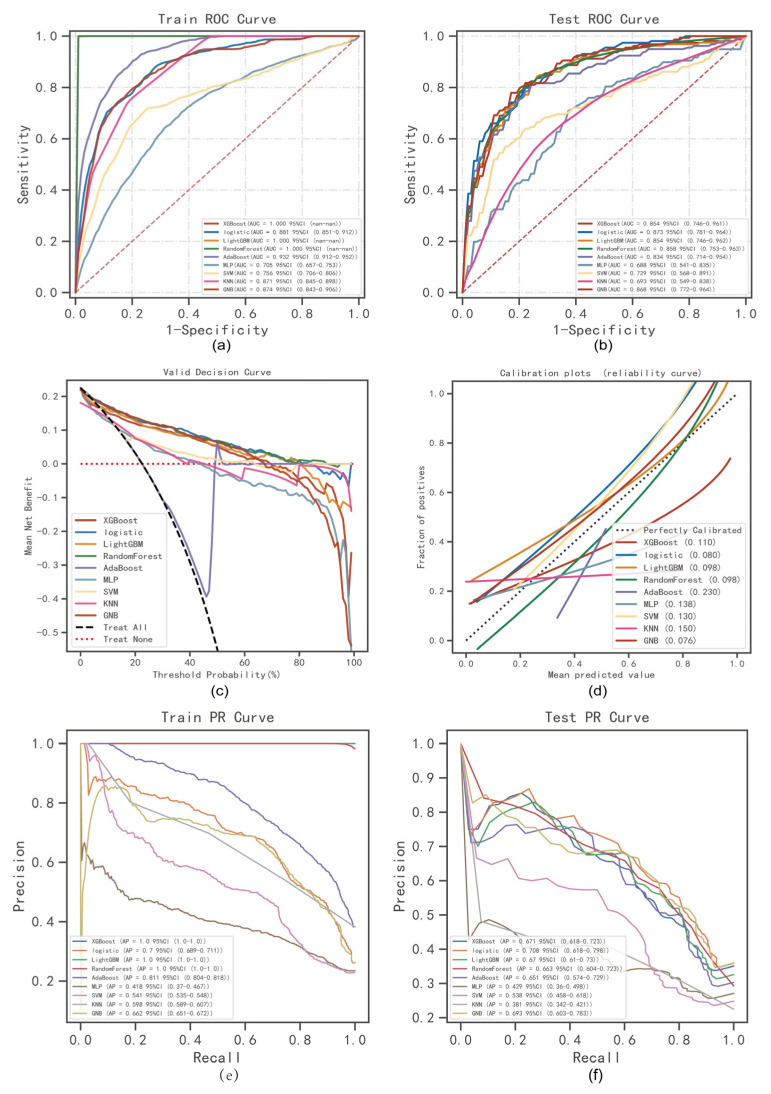
ML model comprehensive analysis. (**a**) Training sets ROC and AUC and (**b**) Testing sets ROC and AUC. Gout patients were sampled 10 times at a ratio of 7:3. (**c**) Test set DCA where the black dotted line represents the assumption that all patients have tophi and the red dotted line and the thin black line represent the assumption that no patient has tophi. The remaining solid lines represent different models. (**d**) For the calibration curve of the test set, the abscissa is the average prediction probability, the case coordinate is the actual probability of the event, the dashed diagonal is the reference line, and the other smooth solid lines are the different model fitting lines. The closer the fitting line is to the reference line, the smaller the value in brackets is, the more accurate the model prediction value is. (**e**) Training set PR curve and AP and (**f**) testing set PR curve and AP. The y-axis is precision and the x-axis is recall. If the PR curve of one model is completely covered by the PR curve of another model, it can be concluded that the latter is better than the former, and the higher the AP value, the better the model performance. The different colors in the picture represent the corresponding model, and the values are represented by the average and 59% CI.

**Figure 3 jcm-12-01755-f003:**
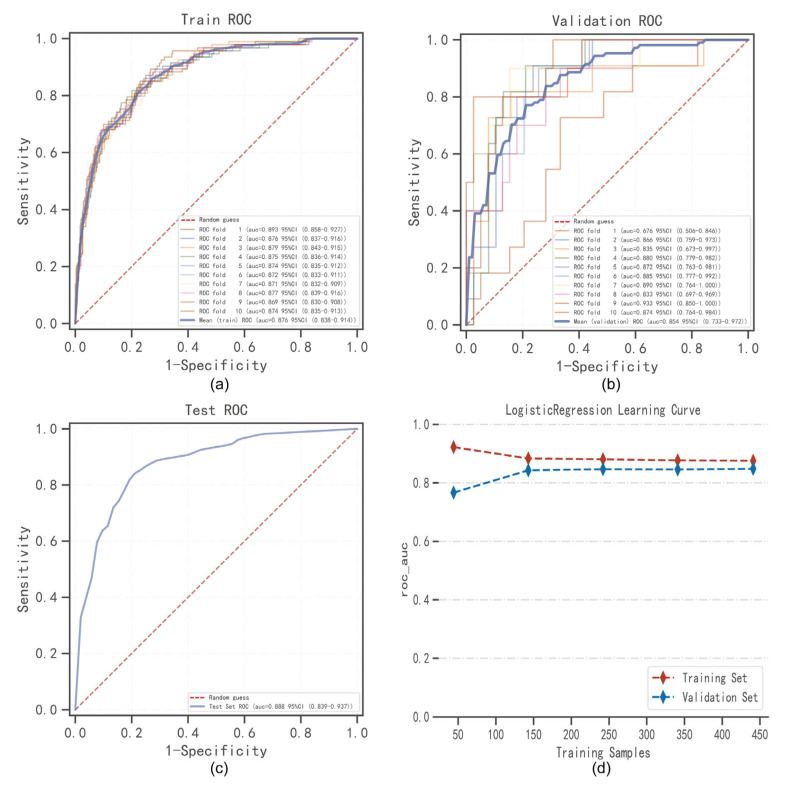
Logistic regression model training, validation, and testing. (**a**) Training sets ROC and AUC and (**b**) validation sets ROC and AUC. Training and cross-validation of 10% of gout patients. Solid lines of different colors represent 10 different results. (**c**) Test set ROC and AUC. Test results for 30% of gout patients. (**d**) Learning curve. The red dashed line represents the training set and the blue dashed line represents the validation set. The values are expressed in terms of average and 59% CI.

**Figure 4 jcm-12-01755-f004:**
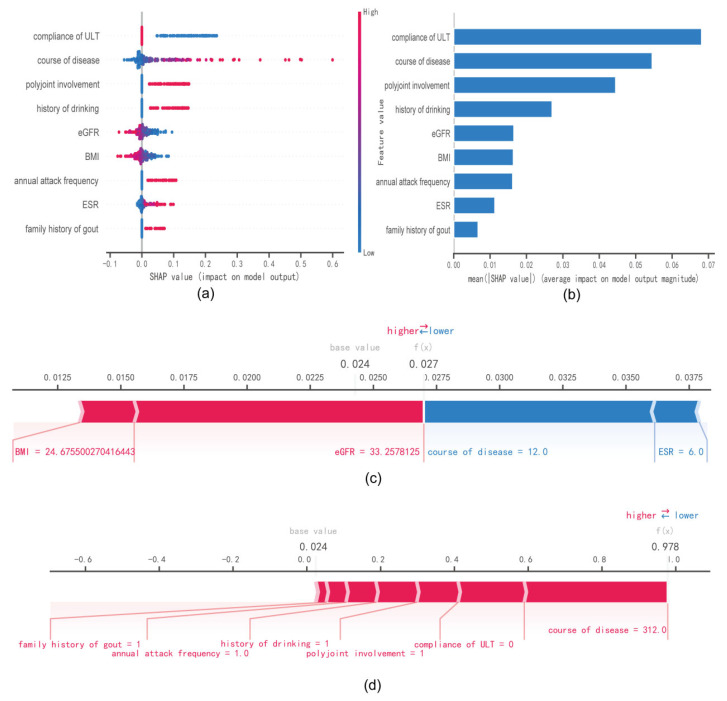
SHAP interprets the model. (**a**) Attributes of characteristics in SHAP. Each line represents a feature, and the abscissa is the SHAP value. Red dots represent higher eigenvalues and blue dots represent lower eigenvalues. (**b**) Feature importance ranking as indicated by SHAP. The matrix diagram describes the importance of each covariate in the development of the final prediction model. (**c**) Individual efforts by patients without tophus and (**d**) with tophus. The SHAP value represents the predictive characteristics of individual patients and the contribution of each to the predictive mortality. The number in bold is the probability forecast value (f(x)), while the base value is the predicted value without providing input to the model. F(x) is the logarithmic ratio of each observation. Red features indicate increased risk of death and blue features indicate reduced risk of death. The length of the arrows helps visualize the extent to which the prediction is affected. The longer the arrow, the greater the effect.

**Table 1 jcm-12-01755-t001:** Baseline characteristics in training cohort and testing cohort.

Variable	Training Set (n = 491)	Testing Set (n = 211)	Z	P
annual attack frequency (times), n (%)				
<6	319 (64.969)	136 (64.455)	1.089	0.58
6–12	74 (15.071)	27 (12.796)		
>12	98 (19.959)	48 (22.749)		
history of smoking, n (%)				
no	224 (45.621)	96 (45.498)	0.001	0.976
yes	267 (54.379)	115 (54.502)		
history of drinking, n (%)				
no	92 (18.737)	46 (21.801)	1.219	0.544
drinking every week < 70 g	269 (54.786)	107 (50.711)		
drinking every week ≥ 70 g/ years of drinking ≥ 10 years	130 (26.477)	58 (27.488)		
history of sugary diet, n (%)				
no	326 (66.395)	151 (71.564)	1.81	0.178
yes	165 (33.605)	60 (28.436)		
history of high purine diet, n (%)				
no	218 (44.399)	80 (37.915)	2.54	0.111
yes	273 (55.601)	131 (62.085)		
history of high altitude residence, n (%)				
no	404 (82.281)	177 (83.886)	0.267	0.606
yes	87 (17.719)	34 (16.114)		
history of hypertension, n (%)				
no	400 (81.466)	159 (75.355)	3.398	0.065
yes	91 (18.534)	52 (24.645)		
history of diabetes, n (%)				
no	474 (96.538)	202 (95.735)	0.267	0.605
yes	17 (3.462)	9 (4.265)		
history of hyperlipidemia, n (%)				
no	311 (63.469)	131 (62.085)	0.121	0.728
yes	179 (36.531)	80 (37.915)		
history of kidney stones, n (%)				
no	376 (76.578)	156 (73.934)	0.562	0.453
yes	115 (23.422)	55 (26.066)		
history of kidney crystallization, n (%)				
no	441 (89.817)	182 (86.256)	1.874	0.171
yes	50 (10.183)	29 (13.744)		
family history of gout, n (%)				
no	404 (82.281)	181 (85.782)	1.302	0.254
yes	87 (17.719)	30 (14.218)		
polyjoint involvement (joints), n (%)				
<3	275 (56.008)	114 (54.028)	0.234	0.628
≥3	216 (43.992)	97 (45.972)		
tophus, n (%)				
no	387 (78.819)	157 (74.408)	1.646	0.199
yes	104 (21.181)	54 (25.592)		
sex, n (%)				
no	8 (1.629)	2 (0.948)	0.488	0.485
yes	483 (98.371)	209 (99.052)		
compliance of ULT, n (%)				
MPR < 60%	230 (46.843)	96 (45.498)	0.107	0.743
MPR ≥ 60%	261 (53.157)	115 (54.502)		
urine specific gravity, median [IQR]	1.014 [1.011, 1.018]	1.015 [1.011, 1.020]	−1.577	0.114
Urine Ph, median [IQR]	6.000 [5.500, 6.000]	5.500 [5.500, 6.000]	0.998	0.304
CysC, median [IQR]	1.010 [0.870, 1.210]	1.020 [0.890, 1.210]	−1.031	0.303
GLOB, median [IQR]	32.400 [29.500, 35.100]	31.500 [29.000, 35.000]	1.293	0.196
ALB, median [IQR]	46.100 [43.600, 48.600]	45.600 [43.500, 48.300]	0.594	0.552
AST, median [IQR]	27.000 [22.000, 35.000]	26.000 [20.000, 34.100]	1.579	0.114
ALT, median [IQR]	32.000 [21.000, 49.600]	31.000 [20.000, 47.000]	0.961	0.337
Crea, median [IQR]	82.500 [73.700, 92.900]	83.500 [74.700, 95.000]	−0.855	0.393
Urea, median [IQR]	4.740 [3.780, 5.700]	4.700 [3.750, 5.840]	0.022	0.983
UA, median [IQR]	527.100 [434.300, 595.900]	507.000 [417.600, 593.300]	1.053	0.293
PDW, median [IQR]	16.500 [16.200, 16.800]	16.500 [16.100, 16.700]	1.882	0.059
PCT, median [IQR]	0.231 [0.202, 0.272]	0.238 [0.199, 0.273]	0.107	0.915
MPV, median [IQR]	11.200 [10.100, 12.600]	11.300 [10.200, 12.700]	−0.446	0.656
PLT, median [IQR]	204.000 [166.000, 249.000]	206.000 [164.000, 245.000]	−0.027	0.978
MCHC, median [IQR]	331.000 [323.000, 337.000]	330.000 [322.000, 339.000]	0.04	0.968
MCV, median [IQR]	92.000 [89.700, 94.400]	91.500 [89.300, 94.500]	0.547	0.584
HCT, median [IQR]	0.462 [0.435, 0.485]	0.456 [0.428, 0.479]	1.844	0.065
HGB, median [IQR]	153.000 [143.000, 161.000]	151.000 [139.000, 160.000]	1.549	0.121
RBC, median [IQR]	5.040 [4.710, 5.350]	4.990 [4.660, 5.340]	1.139	0.255
MO, median [IQR]	0.440 [0.350, 0.580]	0.450 [0.340, 0.580]	−0.138	0.891
LY, median [IQR]	1.930 [1.570, 2.410]	2.000 [1.590, 2.530]	−0.721	0.471
GR, median [IQR]	4.550 [3.690, 6.230]	4.600 [3.420, 6.420]	0.135	0.893
WBC, median [IQR]	7.380 [6.190, 9.090]	7.710 [6.090, 9.260]	−0.371	0.711
ESR, median [IQR]	13.000 [5.000, 26.000]	13.000 [6.000, 29.000]	0.159	0.874
eGFR, median [IQR]	61.149 [49.171, 72.686]	58.768 [47.969, 70.451]	1.541	0.123
course of disease, median [IQR]	48.000 [24.000, 96.000]	60.000 [24.000, 108.000]	−1.171	0.241
BMI, median [IQR]	25.687 [23.459, 27.682]	25.712 [23.437, 27.682]	−0.205	0.838

ULT, urate-lowering therapy; MPR, medication possession ratio; CysC, Cystatin C; GLOB, globulin; ALB, serum albumin; AST, Aspartate aminotransferase; ALT, Alanine aminotransferase; Crea, Creatinine; UA, Uric acid; PDW, Platelet distribution width; PCT, Plateletcrit; MCHC, Mean corpuscular hemoglobin concentration; HCT, Hematocrit; HGB, Hemoglobin; RBC, Red blood cell; MO, Monocyte; LY, Lymphocyte; GR, Granulocte; WBC, White cell rate; ESR, Erythrocyte sedimentation rate; eGFR, estimated glomerular filtration rate; BMI, Body Mass Index.

**Table 2 jcm-12-01755-t002:** Multivariate logistic regression analysis.

Variable	R	SE	Z	*p*	OR (95% CI)
sex	16.646	593.391	0.028	0.978	16,956,319.853 (-)
compliance of ULT	−1.53	0.251	−6.104	<0.001	0.217 (0.131–0.35)
annual attack frequency (>12 times)	0.848	0.273	3.1	0.002	2.334 (1.365–3.996)
annual attack frequency (6–12 times)	−0.156	0.339	−0.461	0.644	0.855 (0.434–1.642)
history of drinking (drinking ≥ 70 g per week/drinking years ≥ 10 years),	0.819	0.33	2.481	0.013	2.268 (1.198–4.386)
history of drinking (drinking < 70 g per week)	−0.632	0.328	−1.929	0.054	0.532 (0.281–1.017)
family history of gout	0.824	0.291	2.83	0.005	2.279 (1.285–4.033)
polyjoint involvement	1.288	0.256	5.028	<0.001	3.624 (2.209–6.041)
course of disease	0.009	0.002	4.858	<0.001	1.009 (1.005–1.012)
BMI	−0.084	0.038	−2.242	0.025	0.919 (0.853–0.989)
ESR	0.013	0.006	2.371	0.018	1.013 (1.002–1.025)
eGFR	−0.024	0.007	−3.333	0.001	0.977 (0.963–0.99)
UA	0.001	0.001	1.515	0.13	1.001 (1–1.1003)
(Intercept)	−16.443	593.392	−0.028	0.978	0 (-)

R, regression coefficient; SE, Standard error; OR, odds ratio; CI, confidence interval; ULT, urate-lowering therapy; BMI, Body Mass Index; ESR, Erythrocyte sedimentation rate; eGFR, estimated glomerular filtration rate; UA, Uric acid.

## Data Availability

Data is contained within the Appendix A. The data presented in this study are available in [Appendix A].

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
