# Peer review of "Establishment and Validation of Predictive Model of Tophus in Gout Patients"

_jcm, 2023, doi:10.3390/jcm12051755_

Round 1

Reviewer 1 Report

The authors present an observational study that aims to explore the factors related to the incidence of tophus and establish a prediction model is clinically significant. A retrospective analysis was conducted on 702 patients with gout, 158 patients with tophus manifestations in comparison with the patients with and without tophus 17 in terms of baseline characteristics. LASSO and multivariate logistic regression were used to analyze predictors. Gout patients were sampled 10 times at a ratio of 7: 3(Training sets and Testing sets). SHAP interprets the model. The study is interesting and the results are imported.

 Comments:

Decreased BMI would elevate the incidence of tophus in gout patients.

Why?

What was the type of the study?

Using the cross-sectional study, data is collected as a whole to study a population at a single point in time to examine the relationship between variables of interest.

Or

Using the retrospective cohort study, the exposure and outcome information (temporality) in a cohort study is identified retrospectively by using administrative datasets.

Generally, the "incidence" term is used in a cohort study.

A possible pathogenesis mechanism, such as decreased BMI associated with tophi in gout patients, would be provided in the discussion section.

Author Response

Dear reviewer,

Thank you for your letter and for the reviewers’ comments concerning our manuscript entitled “Establishment and Validation of Predictive Model of Tophus in Gout Patients”. Those comments are valuable and very helpful for revising and improving our paper. We have studied the comments carefully and have revised the manuscript, according to the comments and suggestions of the reviewers and editor. We respond point-by-point to the comments as listed below, and we hope that the changes we have made in the manuscript will meet with your approval. Edited parts are marked by the “Track Changes” function in the manuscript.

With kindest regards,

Sincerely,

Yu-Feng Qing

The uploaded word file lists the main corrections and responses to the reviewer's comments.

Reviewer 2 Report

Comments

This article explored the factors related to the incidence of tophus and established a prediction model by machine learning. The reviewer thinks the findings of this study supported by the new analytic methodology were little novelty but reasonable for a clinical aspect. The reviewer has some requests for the correction of the manuscript. 

All abbreviations are not explained in the abstract. Please spell them out on the first appearance.

The purpose of the study needs to be clearly stated in either the abstract and the text. Please specify the purpose of this study.

Please correct the scattered hyphens inserted in words.

Regarding the exclusion criteria, why were cardiac, liver, malignant tumor, and psychiatric diseases excluded? How many patients did the authors exclude from this study? What was the prevalence of tophus in the excluded cases? Could these diseases be risk factors? The reviewer would like the authors to answer and add the discussion about this point to the manuscript.

Author Response

(The authors gave the same response as above.)

Round 2

Reviewer 1 Report

No further comment